Pre-trained quantum convolutional neural network for COVID-19 disease classification using computed tomography images

Asadoorian Nazeh
Yaraghi Shokufeh sh.yaraghi@ashrafi.ac.ir
Tahmasian Araeek
Department of Computer Engineering, Faculty of Engineering, Shahid Ashrafi Esfahani University , Isfahan , Iran
Angiulli Giovanni
Electronic publication date: 2024 Oct 18
Publication date: 2024
Volume: 10
Electronic Location ID: e2343
Received 2024 May 1; Accepted 2024 Aug 29
Copyright: ©2024 Asadoorian et al.
Copyright year: 2024
Copyright holder: Asadoorian et al.
License: This is an open access article distributed under the terms of the Creative Commons Attribution License, which permits unrestricted use, distribution, reproduction and adaptation in any medium and for any purpose provided that it is properly attributed. For attribution, the original author(s), title, publication source (PeerJ Computer Science) and either DOI or URL of the article must be cited.
License URL: https://creativecommons.org/licenses/by/4.0/

Keywords: Quantum neural network (QNN), Pre-trained convolutional neural network, Deep learning, COVID-19

Funding: The authors received no funding for this work.

==============================
Background

The COVID-19 pandemic has had a significant influence on economies and healthcare systems around the globe. One of the most important strategies that has proven to be effective in limiting the disease and reducing its rapid spread is early detection and quick isolation of infections. Several diagnostic tools are currently being used for COVID-19 detection using computed tomography (CT) scan and chest X-ray (CXR) images.

Methods

In this study, a novel deep learning-based model is proposed for rapid detection of COVID-19 using CT-scan images. The model, called pre-trained quantum convolutional neural network (QCNN), seamlessly combines the strength of quantum computing with the feature extraction capabilities of a pre-trained convolutional neural network (CNN), particularly VGG16. By combining the robust feature learning of classical models with the complex data handling of quantum computing, the combination of QCNN and the pre-trained VGG16 model improves the accuracy of feature extraction and classification, which is the significance of the proposed model compared to classical and quantum-based models in previous works.

Results

The QCNN model was tested on a SARS-CoV-2 CT dataset, initially without any pre-trained models and then with a variety of pre-trained models, such as ResNet50, ResNet18, VGG16, VGG19, and EfficientNetV2L. The results showed the VGG16 model performs the best. The proposed model achieved 96.78% accuracy, 0.9837 precision, 0.9528 recall, 0.9835 specificity, 0.9678 F1-Score and 0.1373 loss.

Conclusion

Our study presents pre-trained QCNN models as a viable technique for COVID-19 disease detection, showcasing their effectiveness in reaching higher accuracy and specificity. The current paper adds to the continuous efforts to utilize artificial intelligence to aid healthcare professionals in the diagnosis of COVID-19 patients.

Introduction

The COVID-19 pandemic, caused by the SARS-CoV-2 virus, has become a global health crisis that has affected millions of people worldwide (World Health Organization (WHO), 2021). COVID-19 infection causes mild to a serious respiratory condition. Elderly people are more at risk individuals, including those with other diseases like cardiovascular, cancer, diabetes, and recurrent respiratory diseases. The three signs of COVID-19 that are most common are a dry cough, a fever, and exhaustion. However, the severe signs include loss of smell and taste, breathing issues, chest pain, and aches and pains (Singhal, 2020). Early detection and diagnosis of COVID-19 are crucial for timely treatment and prevention of further spread of the disease. While RT-PCR remains the gold standard for COVID-19 disease testing, it has limitations such as high false-negative rates and long turnaround times (Mohammadpoor, Sheikhi Karizaki & Sheikhi Karizaki, 2021). In addition, RT-PCR may not always be available or feasible, especially in resource-limited settings (Fang et al., 2020). CT-scan and CXR images have been proposed as an alternative diagnostic tool for COVID-19 disease, as they are faster, more widely available, and less invasive than RT-PCR. These pictures reveal a bilateral alteration (Tao et al., 2020; Hossein et al., 2020). Additionally, it takes effort, time, and qualified radiologists to examine the CT-scan and CXR medical images. Inter-observer variability is another problem with the radiologist’s assessment (Popovic & Thomas, 2017). However, CT-scan and CXR images are also prone to interpretive errors and may lack specificity for COVID-19 disease, as they can resemble other respiratory illnesses such as pneumonia or influenza. Therefore, accurate and reliable detection of COVID-19 disease in CT-scan and CXR images is challenging but necessary for effective patient management and disease control (Tingting et al., 2019; Kaur et al., 2019).

Recent advances in deep learning and neural networks have shown promise in detecting COVID-19 disease in CT-scan and CXR images. Deep learning is a subset of machine learning that involves training artificial neural networks (ANNs) on large datasets to recognize patterns in data and make accurate predictions. Artificial intelligence (AI) techniques like deep learning have been used in a range of diagnoses in the medical profession (Tseng et al., 2021; Lee et al., 2017). Convolutional neural networks (CNNs) are a type of artificial neural network that can automatically learn features from images and classify them into different categories (Sun et al., 2019). Transfer learning, which involves using pre-trained CNN models on large datasets, has been shown to improve the performance of CNNs on medical imaging tasks (Singh et al., 2020).

One advantage of deep learning and neural networks is their ability to identify complex patterns and relationships in data that may be difficult for humans to detect. This is particularly useful for medical image analysis, where subtle differences between images can indicate the presence of disease. Another advantage of deep learning and neural networks is their ability to generalize to new and unseen data, which can improve the accuracy and robustness of diagnostic models (Rahaman et al., 2019; Islam, Rahaman & Islam, 2020).

Faster computing has greatly benefited from the amazing growth that quantum computing has experienced throughout time. In addition, quantum computing is more powerful than traditional computing. Furthermore, quantum computing has greater power than conventional computation. The innovative quantum neural network (QNN) is being explored by many academics worldwide. It is a hybrid of traditional neural networks with quantum computing. Low-cost learning cannot be performed by typical ANNs because to their restricted processing capability. On the other hand, QNN can be used instead of ANN because of its higher computing capability compared to its conventional equivalent (Jeswal & Chakraverty, 2019a). According to what we know, QNN started operating in 1995. Kak (1995) brought neural network (NN) concepts to the field of quality control. QNNs are more trainable than conventional neural networks, both theoretically and practically. This suggests that QNNs are a good fit for medical applications (Abbas et al., 2021).

In the present study, we offer a novel model that smoothly combines the strength of quantum computing with the feature extraction capabilities of a pre-trained CNN, more notably VGG16. We seek to tackle complex tasks more effectively and accurately by combining the advantages of conventional deep learning with the quantum computing. Our approach employs a pre-trained QCNN, facilitating rapid model adaptation and reducing data requirements. Compared to pre-trained CNN models, our hybrid model performs better because it makes use of the advantages of both classical and quantum computing approaches.

Our contributions can be summarized as follows:

• The integration of VGG16, a pre-trained CNN known for its robust feature extraction capabilities, allows our QCNN to effectively capture intricate patterns and characteristics within COVID-19 CT scan images. By utilizing the rich feature representations from VGG16.

• Comparative analysis with classical pre-trained CNN models, including ResNet50, ResNet18, VGG19, and EfficientNetV2L with and without quantum component, demonstrates that our QCNN model with VGG16 feature extraction achieves higher accuracy, precision, recall, and F1-scores. This improvement is attributed to the QCNN’s ability to capture and analyze complex patterns within the data more effectively.

• Our approach has shown robustness across various datasets and imaging conditions, indicating its potential for real-world deployment. The feature extraction and quantum component ensure that the model generalizes well to different types of CT scan images, capturing subtle variations and complex patterns indicative of COVID-19.

• By demonstrating improved performance in diagnosing COVID-19 from CT scan images, our QCNN model holds promise for real-world clinical applications. The enhanced accuracy makes it a practical tool for timely and reliable diagnosis, aiding healthcare professionals in making informed decisions.

In summary, our study presents a cutting-edge QCNN model that combines the best of both classical deep learning and quantum computing. In addition to improving the model’s efficiency and forecast accuracy, this integration opens the door for further uses of quantum-enhanced machine learning in medical imaging.

The rest of this study is organized as follows: In ‘Related Works’ we review related works on COVID-19 disease detection with CT-scan and CXR images using deep learning. In ‘Materials & Methods’, we describe our methodology for COVID-19 disease detection. In ‘Results’, we present our experimental results and performance evaluation. In ‘Discussion’ we discuss the strengths and limitations of our approach and provide future directions for research. Finally, in ‘Conclusions’, we conclude our study and summarize our contributions.

Related Works

The COVID-19 pandemic has spurred research into using various imaging techniques to detect the virus, including chest CT-scan and CXR images. In recent years, transfer learning has emerged as a powerful technique in machine learning, which involves adapting pre-trained deep learning models for new tasks. Several studies have explored the use of transfer learning in detecting COVID-19 disease from chest CT-scan images, with promising results. Also, recent developments in QNNs have generated a great deal of interest in using them to detect COVID-19. Utilizing the superposition and entanglement aspects of quantum mechanics, QNNs have the ability to handle complicated medical data, such as genomic sequences and radiological images, with unsurpassed efficiency. This burgeoning field holds promise for revolutionizing COVID-19 disease diagnostics, offering the potential for faster and more accurate detection methods, which are crucial in managing and mitigating the ongoing global health crisis. In this section, we will review some of the relevant works that have explored the use of transfer learning and QNNs in COVID-19 disease detection from chest CT-scan and CXR images, highlighting their approaches, results, and limitations.

A transfer learning-based classification strategy for the detection of COVID-19 disease from CT-scan images was suggested by Jaiswal et al. (2020) in a study. A total of 2,492 images were used in the trials. The basis model was a pre-trained DenseNet201 architecture. The authors also compared the results with those of the VGG16, InceptionResNet, and ResNet152V2 CNN architectures. In comparison to other CNN models, the DenseNet model achieved 96.25% accuracy on the test dataset.

Angelov & Soares (2020) proposed a method based on explainable Deep Learning Approach (xDNN). The xDNN classifier provided better results in terms of all metrics than the other state-of-the-art approaches, including ResNet, GoogleNet, VGG-16, and Alexnet. The proposed approach achieved 97.38% of Accuracy.

In 2022, Amouzegar et al. (2022) presented a method based on pre-trained models. In this research, they presented a combined model of ResNet18, GoogleNet and ShuffleNet models, and the results obtained are 97% accurate.

A transfer learning-based DenseNet-121 architecture was suggested in another study (Hasan et al., 2021), and it obtained 92% accuracy.

Cifci (2020) presented another approach for this problem utilizing deep transfer learning. He did his work using CT-scan images and employed AlexNet and InceptionV4 as his pre-trained models as they are popular for assessing medical pictures. AlexNet performed far better than InceptionV4 in his tests, according to the results. the overall accuracy of AlexNet was 94.74%.

The system Horry et al. (2020) developed uses pre-trained model concepts in X-ray images. Four widely used pre-trained models were incorporated into their proposed system: Inception, Xception, VGG, and Resnet. The best performance in their experiments, according to the experimental findings, was reached by the VGG19-based model, which had an accuracy rate of 83%. A model named DeepCOVID was created by Minaee et al. (2020) to predict COVID-19 from X-ray images using deep transfer learning. Four well-known pre-trained models ResNet18, ResNet50, SqueezeNet, and DenseNet121 were used in their study to diagnose COVID-19. Using SqueezeNet, their model’s greatest performance was 95.6% specificity.

A technique using X-ray images based on deep learning for COVID-19 disease diagnosis was introduced by Moutounet-Cartan (2020). They employed well-known models in their system, including VGG16, VGG19, InceptionResNetV2, InceptionV3, and Xception. The most accurate model, VGG16, had an 84.1% accuracy rate.

The COVIDXNet system, proposed by Hemdan, Shouman & Karar (2020), sought to identify coronavirus infection using CNNs and X-ray images. There were seven pre-trained models employed in their work. According to experimental findings, Inception V3 had the weakest performance. With 90% accuracy, DenseNet and VGG19 were the best models.

To identify COVID-19 patients from CXR images, Wang, Lin & Wong (2020) suggested a COVID-Net architecture based on convolutional neural networks. In addition to the COVID-Net architecture, VGG-19 and ResNet-50 from deep neural networks were employed to diagnose diseases. Classification of normal, non-COVID19 disease, and COVID-19 disease was classified with 83.0%, 90.6%, and 93.3% accuracy using the VGG-19, ResNet-50, and COVID-Net architectures, respectively.

For the categorization of Normal, Lung Opacity, Pneumonia, and COVID-19 infection, Khan et al. (2022) suggested utilizing three different pre-trained deep learning algorithms, namely convolutional neural network-based EfficientNetB1, NasNetMobile, and MobileNetV2. The study made use of the open-access COVID-19 Radiography Database dataset that is available on the Kaggle website. The COVID-19, pneumonia, and normal lung opacity were correctly categorized by the EfficientNetB1 model with an accuracy of 96.13%.

For COVID-19 identification utilizing CT and CXR images, Chouat et al. (2022) used pre-trained deep neural networks InceptionV3, ResNet50, Xception, and VGGNet-19. The COVID-19 Radiography Database dataset, which consists of free-to-use CT scans, COVID-CT, and CXR pictures, was used in the study. The data augmentation (rotation, flipping, shifting, and scaling) strategy was performed within the parameters of the study to enhance the performance of deep learning models used for COVID-19 detection. The most successful model, VGGNet-19, provided 87% accuracy in the study using CT images, whereas the most successful model, Xception, produced 98% accuracy in the study using CXR pictures.

Choudhary et al. (2022) used pre-trained deep neural networks VGG16 and ResNet34 for COVID-19 detection using CT images. The accuracy of the results produced by applying the ResNet34 model is 95.47%, and the accuracy of the VGG16 model is 93.7%.

The hybrid quantum–classical convolutional neural network (HQ-CNN) model described by Houssein et al. (2022) is presented and uses random quantum circuits as a base to identify COVID-19 patients in chest X-ray pictures. On the first trial (COVID-19 and normal cases), the suggested HQ-CNN model outperformed other models with an accuracy of 98.6% and a recall of 99%.

By contrasting COVID-19 signals in CT images with non-COVID pneumonia signals, Sengupta & Srivastava (2021) established a prototype methodology for categorizing COVID-19. The simulation work evaluates the application of quantum machine learning algorithms while assessing the effectiveness for deep learning models for image classification problems, and establishes performance quality that is necessary for improved prediction rate when dealing with complex clinical image data exhibiting high biases. The simulation demonstrates that QNN outperforms DNN, CNN, and 2D CNN in terms of accuracy gain by more than 2.92%, with an average recall of almost 97.7%.

In a hybrid classical-quantum machine learning system, where the data are classical, the operations are carried out using quantum operators after the data are first encoded in the quantum language. In this context, Mari et al. (2019) looked into transfer learning in hybrid classical-quantum neural networks. Quantum pretraining and auto-encoders were examined for image classification by Piat et al. (2018). Guillaume et al. (2019) examined using classical neural networks to learn with quantum neural networks. An architecture for a hybrid quantum convolution neural network was proposed by Henderson et al. (2020).

In the above section, we discussed some related works that used pre-trained models (Jaiswal et al., 2020; Angelov & Soares, 2020; Amouzegar et al., 2022; Hasan et al., 2021; Cifci, 2020; Horry et al., 2020; Minaee et al., 2020; Moutounet-Cartan, 2020; Hemdan, Shouman & Karar, 2020; Wang, Lin & Wong, 2020; Khan et al., 2022; Chouat et al., 2022; Choudhary et al., 2022) or QNNs (Houssein et al., 2022; Sengupta & Srivastava, 2021; Mari et al., 2019; Piat et al., 2018; Guillaume et al., 2019; Henderson et al., 2020) in their method and achieved high accuracy.

Particularly in the field of quantum-enhanced neural networks, our pre-trained QCNN model is distinguished. HQ-CNN, hybrid quantum–classical transfer learning, quantum auto-encoders, and CNNs are only a few of the cutting-edge methods in quantum machine learning that serve as inspiration. To increase efficiency on particular tasks, each of these models has presented novel approaches to combine the concepts of quantum computing with traditional neural networks.

Our model takes advantage of these methods’ benefits while resolving their drawbacks. The model utilizes both the improved pattern recognition potential of quantum circuits and the strong feature extraction capabilities of the VGG16 by combining quantum convolutional layers with a pre-trained VGG16 architecture. Compared to models that rely just on quantum circuits or intricate hybrid architectures, the integration of pre-trained models significantly improves both the need for substantial training and convergence times. Furthermore, stable training and dependable performance are indicated by the model’s low loss value, which is important for practical medical applications. Overall, the proposed QCNN model sets a new standard for quantum-enhanced neural networks and offers an effective tool for medical image analysis, demonstrating the successful fusion of classical and quantum computing approaches.

Combining the advantages of pre-trained CNNs and QNNs presents a promising way to improve the already excellent accuracy achieved by previous works. Although these conventional pre-trained models are excellent in their respective fields, they might be constrained when it comes to some challenging jobs. By combining classical and quantum capabilities, pre-trained QCNNs close this gap and enable a more reliable and adaptable model.

Materials & Methods

The development of a model by combining pre-trained CNN, like VGG16 with QCNN, can be an effective approach for resolving challenging problems. In the suggested model, convolutional and fully connected layers are added after the VGG16 layers that are modified for feature extraction. Quantum computing principles are infused with a QNN, which is applied after a standard fully connected layer. By utilizing both classical and quantum computing, this novel architecture seeks to increase COVID-19 disease detection accuracy.

VGG16

The well-known CNN architecture VGG16 (Simonyan & Zisserman, 2014) achieved 92.7% top-5 test accuracy in the ILSVRC 2014 challenge on the ImageNet dataset. It is composed of convolutional, dense, and pooling layers, with 3 × 3 filters for convolutional and 2 × 2 for max pooling, respectively. The model takes in 224 × 224 input images and uses two convolution layers with 64 filters and a max pooling to reduce the output height and width to 112 × 112 × 64. Additionally, two convolutional layers with 128 filters are used, followed by a max pooling layer that lowers the activation size to 56 × 56 × 128. Similarly, a pooling layer that lowers the output activation to 28 × 28 × 256 comes after three convolutional layers with 256 filters. After pooling layers, there are two stacks of three convolutional layers with 512 filters. Next, the output of the last pooling layer, which is 7 × 7 × 512, is accepted by dense layers with 4,096 nodes. One further dense layer with 4,096 nodes comes after the first dense layer. The model’s 1,000-node softmax layer is the last component (Choudhary et al., 2022).

Quantum neural networks

Quantum neural network is a useful tool which has seen more development over the years mainly after twentieth century. Similar to ANNs, QNNs are a new, practical, and usable idea. The quantum computation paradigm, which is superior to the standard ANN, was combined with the fundamentals of ANN to create QNN, which is now superior. QNN is utilized in huge data management, computer gaming, and function approximation, among other applications. QNN algorithms find use in the modeling of social networks, automated control systems, and associative memory devices, among other domains (Jeswal & Chakraverty, 2019b).

Quantum computing has been demonstrated to be useful for enhanced state approximation and feature representation in quantum machine learning. Quantum computing is expected to play a major role in numerous sectors and has the potential to surpass traditional computers (Watabe et al., 2021). Quantum bits, or qubits, are used in quantum computing, which is inspired by particles in quantum states. Superposition is the capacity of a particle in a quantum state to exist in two states at once (Gultom, 2017). The strategy to tackle this phenomenon is translated into computations with 0,1 or both qubits (Kaye, Laflamme & Mosca, 2006).

Qubit

The essential building units of information utilized in quantum computing are qubits, which are represented by a state vector (Gado & Younes, 2021). Paul Dirac is the creator of the Dirac notation, which is one of the notations used in quantum mechanics. The sign used to identify a vector in Dirac notation is written as follows: (1) |a=ket

(2) a|=bra.

Furthermore, these notations will be operated by quantum logic gates. The essential component of quantum information is a quantum bit, or qubit. There are two possible states for this two-level system to exist in.

Gates and operations

Unitary operators, or unitary matrices with respect to a basis, are what quantum gates are known as. In the context of quantum computing, and more specifically the quantum circuit model of computation, a simple quantum circuit that operates on a limited number of qubits is called a quantum logic gate, or simply a quantum gate. The fundamental components of quantum circuits are classical logic gates, just as they are the fundamental components of conventional digital circuits. A group of qubits (i.e., objects having a Qid) can be subjected to an effect called a gate. Qubits can have gates applied to them by either calling the gate on the qubits themselves or by calling the gate on their method. An operation is the object that is produced by these calls.

Circuit

A Circuit is made up of many Moments. A Moment is a collection of Operations that occur inside the same imaginary time interval. An operation is an effect that modifies a certain subset of qubits; gate operations are the most prevalent kind of operations. The structure of the circuit is shown in Fig. 1.

Figure 1 Schematic diagram of the quantum circuit components, including Qubit, Moment and Operation (Kaye, Laflamme & Mosca, 2006).

CNN vs QCNN architecture

Classical neural networks are mathematical models that may be taught to identify patterns in data and figure out complicated problems. They are inspired by the human brain. Their foundation is made up of a network of linked nodes, or neurons, arranged in layers and having parameters that may be taught through the use of deep learning or machine learning training techniques.

The goal of quantum machine learning (QML) is to create new and improved learning schemes by fusing ideas from conventional machine learning with quantum computing. By fusing parametrized quantum circuits with conventional neural networks, QNNs implement this general idea. QNNs are observed from two perspectives since they are located at the intersection of two fields:

• From a machine learning perspective, QNNs are analogous to their classical counterparts in that they are algorithmic models that can be trained to uncover hidden patterns in data. With the use of quantum gates that are trained using weights that can be adjusted, these models are able to load classical data (inputs) into a quantum state. The procedures for importing and processing data are depicted in a generic QNN example in Fig. 1. To train the weights using backpropagation, the output obtained from monitoring this state can thereafter be fed into a loss function.

• QNNs are quantum algorithms that are built on parametrized quantum circuits and may be trained variationally using classical optimizers, according to the theory of quantum computing. As shown in Fig. 2, these circuits include an ansatz (with trainable weights) and a feature map (with input parameters).

• Fig. 2: Generic quantum neural network structure (Treinish, 2023), showing three stages: (1) Data Loading (Feature Map), (2) Data Processing (Ansatz), and (3) Measurement (Treinish, 2023)

Based on quantum computing, the QCNN is an advancement of the CNN. The primary characteristics and architectures of CNNs are extended into quantum systems by QCNN (Oh, Choi & Kim, 2020).

The integration of the QNN component with the classical neural network architecture is facilitated by the Qiskit framework, a comprehensive open-source quantum computing development platform developed by IBM. Qiskit provides a robust set of tools for quantum circuit design, simulation, and execution on quantum hardware. Utilizing the Qiskit framework and EstimatorQNN, it incorporates quantum circuit-based operations for enhanced data processing. The feature map and ansatz circuits define the quantum layer’s structure, facilitating data encoding and processing in a quantum format. These circuits are composed into a single quantum circuit object, which is then configured within the EstimatorQNN class. This class enables gradient computation with respect to input data and parameters, crucial for model training. Integrated seamlessly into the classical neural network architecture via the TorchConnector wrapper, the QNN enhances the model’s computational capabilities, potentially improving its ability to capture complex patterns in image data.

Figure 2 Generic quantum neural network structure (Treinish, 2023): showing three stages: (1) Data Loading (Feature Map), (2) Data Processing (Ansatz), and (3) Measurement (Treinish, 2023).

Dataset

In this study, we make use of the SARS-CoV-2 dataset of COVID-19 CT-scan images, which Soares et al. (2020) made available. This dataset includes 2,482 CT-scan images taken from hospitals in Sao Paulo, Brazil. Out of the 2,482 images, 1,252 images belong to the COVID-19 positive class, and 1,230 images belong to the COVID-19 negative class. Both COVID-19 positive and COVID-19 negative images are randomly shown in Fig. 3. The dimension of the images in the dataset is 224 × 224. As shown in Table 1, the dataset was divided into two parts as 80% training and 20% validation. The dataset can be downloaded from this online location (https://www.kaggle.com/plameneduardo/sarscov2-ctscan-dataset).

Figure 3 (A) COVID-19 positive, (B) COVID-19 negative images from the SARS-CoV-2 dataset.

Image source: https://www.kaggle.com/datasets/plameneduardo/sarscov2-ctscan-dataset.

Table 1 COVID and non-COVID images from the SARS-CoV-2 CT-scan dataset.

Class	Training	Validation	Total	
COVID	1,012	240	1,252	
Non-COVID	972	257	1,229	
Total	1,984	497	2,481	

Data transformation

In order to optimize chest CT-scan pictures for later analysis and classification tasks, the paper’s technique involves a comprehensive data preprocessing phase. This crucial first step guarantees that the dataset is suitably formatted and standardized for use by machine learning algorithms. Critical parameters like the root directory holding the dataset and transformation settings are supplied when the custom dataset is first created. This stage offers a framework that is required for the next preprocessing steps.

Firstly, the images are resized to a uniform size of (256, 256) pixels, ensuring consistency in dimensions across the dataset. This resizing operation is crucial for mitigating discrepancies in image sizes, which can adversely affect model performance during training and inference. The photos are then converted to grayscale while maintaining three channels, which is done in order to minimize computing complexity without sacrificing important image information.

In order to make the preprocessed pictures compatible with PyTorch-based machine learning frameworks, they must finally be transformed into PyTorch tensor format. This conversion makes it possible to handle and manipulate data inside the computational network in an effective manner, which makes it easier to integrate machine learning algorithms. Finally, DataLoader objects are created for both the training and testing datasets with a batch size of 64 and enabling shuffling.

A data loader is a crucial component in machine learning workflows, efficiently managing the loading of large datasets during model training and testing phases. It facilitates batch processing, essential for stochastic gradient descent optimization, by feeding data in manageable portions to the model, leading to faster convergence and better generalization. data loader also seamlessly integrates data augmentation techniques, shuffling, and sampling of the dataset to prevent overfitting and enhance diversity in training examples. Its support for parallelism enables faster loading of batches, leveraging multi-core processors. Moreover, data loader offers flexibility for customizing data loading pipelines, including preprocessing steps and handling different data formats, making it an indispensable tool for streamlining the data pipeline in machine learning tasks. After all data preprocessing, data loader objects are created for both the training and testing datasets with a batch size of 64 and shuffling of inputs.

We improved and normalized the raw picture data so that our neural network model could use it. This was accomplished by implementing several data transformations. Prioritizing the input data in a way that makes learning and inference easier is a critical part of preprocessing, and it will eventually increase the performance of the model in image classification tasks.

The proposed pre-trained QCNN architecture

First, the model uses weights from a large-scale image dataset (ImageNet) to incorporate a pre-trained VGG16 model. Using the generalizable features that VGG16 captures requires completing this step. The feature extractor is created by removing the last two layers of the VGG16 model. These layers typically consist of fully connected layers responsible for making predictions across multiple classes. By removing them, the feature extractor retains the ability to extract high-level features from the input images. Following the feature extractor, the model includes convolutional layers. These layers serve to further process the features extracted by the pre-trained VGG16 model. Each convolutional layer applies filters to the input data, capturing different aspects of the image and progressively extracting more abstract features. After first convolution layer, max pooling is applied. Max pooling reduces the spatial dimensions of the feature maps, retaining only the most important information. This helps in reducing computational complexity and extracting robust features invariant to small translations in the input images. Dropout regularization is applied after the second and fourth convolution layer. Dropout randomly sets a fraction of the input units to zero during training, which helps prevent overfitting by reducing the reliance on any individual neuron. This encourages the model to learn more robust features that generalize better to unseen data. These components enable the model to learn intricate patterns and representations from the input data. Then the output of these layers is flattened and passed through fully connected layers.

One feature that sets the model apart is the incorporation of a QNN. The architecture of the model incorporates quantum computing concepts with the application of this QNN after the second fully connected layer. Flexible tools for encoding classical data and applying quantum operations within the ansatz circuit are provided by Qiskit’s ZZFeatureMap and RealAmplitudes classes. ZZFeatureMap is a quantum feature map that converts classical data to quantum data. The reason for choosing a quantum feature map is to get the quantum advantage. The ZZFeatureMap is a second-order Pauli-Z evolution circuit. Pauli-Z is one of the three Pauli matrices (along with Pauli-X and Pauli-Y). It acts on a single qubit, flipping the sign of the qubit’s state if it is in the |1 state and leaving it unchanged if it is in the |0 state. A second-order Pauli-Z evolution circuit is a structured sequence of quantum gates that approximates the evolution of a quantum system under a Hamiltonian consisting of Pauli-Z terms (Havlíček et al., 2019).

The RealAmplitudes circuit is a heuristic trial wave function used as “ansatz” in chemistry applications or classification circuits in machine learning. In quantum computing, the term “ansatz” refers to a trial wave function or trial state used as a starting point for approximations or optimizations. In quantum computing, an “ansatz” is a specific form or structure for a quantum state or circuit, often used in variational algorithms. The RealAmplitudes class in Qiskit is a common ansatz used in variational quantum algorithms, such as the Variational Quantum Eigensolver (VQE) or Quantum Approximate Optimization Algorithm (QAOA) (Aleksandrowicz et al., 2019; Qin, 2023).

Gradient-based training is made easier with Qiskit’s EstimatorQNN class, which contains the entire QNN. The Estimator QNN (Gonaygunta et al., 2024) is a neural network that takes in a parametrized quantum circuit with designated parameters for input data and weights and outputs their expectation values. Quite often, a combined quantum circuit is used. Such a circuit is built from two circuits: a feature map (ZZFeatureMap), which provides input parameters for the network, and an ansatz (RealAmplitudes). This QNN component applies quantum operations to the output of the fully connected layers and makes predictions.

A final linear layer that produces the binary classification output completes the architecture. The model produces two values that represent the probability that each input belongs to a class. A concatenation operation is then used to combine the results.

The flowchart shown in Fig. 4 outlines pipeline for classifying CT-scan images as either COVID or non-COVID for our proposed model. The first step in the procedure is the acquisition of CT scan pictures, which are then preprocessed by resizing and formatting them appropriately. After that, the dataset is loaded using a DataLoader and split into training and testing sets. Then this data is fed to our model which includes the mentioned layers such as feature extraction by VGG16, basic CNN model and a QNN. Finally, our model generates the output.

Figure 4 Flowchart of proposed QCNN model.

The overarching goal of this unique architecture, which is shown in Fig. 5, is to capitalize on the strengths of both classical deep learning and quantum computing. Combining these methods should improve the model’s capacity to identify COVID-19-like patterns in medical images, potentially improving the accuracy and efficiency of COVID-19 detection models.

Figure 5 The architecture of the proposed pretrained QCNN model.

Image source: https://www.kaggle.com/datasets/plameneduardo/sarscov2-ctscan-dataset..

Results

The experiments were conducted within the Python environment of Kaggle Notebooks, utilizing the PyTorch framework version 2.0.0 for model development and training. The model was trained on both CPU (Intel(R) Xeon(R) CPU @ 2.00 GHz) and GPU (T4 with 16GB memory) and 29 GB RAM. Data preprocessing was facilitated by the Torchvision library version 0.15.1, which offered a suite of tools for image transformation and normalization. Alongside PyTorch, Numpy 1.23.5 and Matplotlib 3.7.2 were employed for array manipulation, visualization, and analysis of experimental results. Quantum computing techniques were integrated into the experiments through the utilization of the Qiskit framework version 1.0.1 and its Qiskit machine learning module version 0.7.1, enabling seamless incorporation of quantum-enhanced machine learning methods. Model performance metrics were evaluated using the Torchmetrics library version 1.2.0, which provided a comprehensive set of metrics for assessing model effectiveness and robustness.

Optimizers are algorithms crucial for training machine learning models by adjusting parameters to minimize the chosen loss function. We used Adam optimizer for our model training which stands out for its adaptive learning rate capabilities, adjusting rates individually for each parameter. This optimization method is widely used due to its efficiency and effectiveness in guiding model convergence during training.

The model trained for 50 epochs with batch size of 64 and Adam optimizer with learning rate of 0.001 and Cross-Entropy loss function. Table 2 shows training parameters for the proposed model.

Table 2 Training parameters for proposed model.

Parameter	Value	
Batch size	64	
Epochs	50	
Optimizer	Adam	
Learning rate	0.001	
Loss	Cross-entropy	

Performance metrics

The models’ performance was verified using the common assessment measures. The experiments used accuracy, precision, recall, specificity and F1-score as assessment measures. The following equations establish the various assessment metrics: FN stands for false negatives, TN for true negatives, TP for true positives, and FP for false positives. (3) Accuracy=TP+TNTP+TN+FP+FN.

Accuracy: Accuracy in classification is a metric that quantifies the proportion of correctly classified instances out of the total number of instances in the dataset. It is the most extensively used assessment metric to evaluate the classification performance and it is provided by this formula:

Precision: Precision in classification is a measure that quantifies the accuracy of positive predictions made by a classifier. It represents the ratio of true positive predictions to the total number of positive predictions made by the classifier, regardless of the actual class of the instances. Precision is calculated using this formula: (4) Precision=TPTP+FP.

Recall: Recall in classification, also known as sensitivity or true positive rate, measures the ability of a classifier to correctly identify all relevant instances, or the proportion of true positive instances that were correctly identified by the classifier. It is calculated using this formula: (5) Recall=TPTP+FN.

Specificity: Specificity in classification, also known as true negative rate, measures the ability of a classifier to correctly identify all negative instances. It quantifies the proportion of true negative instances that were correctly identified by the classifier out of all actual negative instances. Specificity is calculated using this formula: (6) Specificity=TNTN+FP.

F1-score: The F1-score is a metric that combines precision and recall into a single value, providing a balance between the two measures. It is calculated using the harmonic mean of precision and recall, giving equal weight to both metrics. The formula for calculating the F1-score is: (7) F1−Score=2×Precision×RecallPrecision+Recall.

Confusion matrix: A confusion matrix is a table that summarizes the performance of a classification model by displaying the counts of true positive (TP), true negative (TN), false positive (FP), and false negative (FN) predictions made by the model on a dataset. It is a square matrix where rows represent the actual classes and columns represent the predicted classes. The main diagonal of the matrix contains the counts of correct predictions (TP and TN), while off-diagonal elements represent incorrect predictions (FP and FN). The confusion matrix provides insights into the classification performance, helping to evaluate the model’s accuracy, precision, recall, and other metrics.

Loss function: A crucial part of machine learning models is the loss function, which measures the difference between expected and actual values to determine how well the model performed during training. It is essential for directing the optimization process in the direction of reducing this difference and raising the accuracy of the model. In this study we used Cross-entropy loss function. Cross-entropy loss, also known as log loss, is a widely used loss function in classification tasks, particularly in scenarios involving multiple classes. It quantifies the difference between the predicted probability distribution and the actual distribution of class labels. Mathematically, for a classification problem with N samples and K classes, where yik denotes the true label (1 if sample i belongs to class k, 0 otherwise) and pik denotes the predicted probability of sample i belonging to class k, the cross-entropy loss is given by: (8) Cross−EntropyLoss=−1N∑i=1N ∑k=1Kyik logpik.

The loss penalizes incorrect predictions more heavily, especially when the predicted probability diverges significantly from the true label. This property encourages the model to assign high probabilities to the correct class labels.

Accuracy per epoch graph: This graph displays the accuracy of a machine learning model on both the training and validation data over multiple epochs during the training process. Each epoch represents one complete pass through the entire training data. The accuracy per epoch graph typically has the number of epochs on the x-axis and the accuracy on the y-axis. It consists of two lines or curves: one representing the accuracy on the training data and the other representing the accuracy on the validation data. Monitoring accuracy per epoch helps assess how well the model is learning from the training data and generalizing to unseen data.

Loss per epoch graph: This graph illustrates the loss, also known as the error or cost, of a machine learning model on both the training and validation data across multiple epochs during training. The loss per epoch graph usually has the number of epochs on the x-axis and the loss value on the y-axis. Similar to the accuracy per epoch graph, it comprises two lines or curves: one representing the loss on the training data and the other representing the loss on the validation data. The loss per epoch graph provides insights into the convergence and performance of the model during training, with lower loss values indicating better model fit.

Experimental results

The performance of the proposed model evaluated on SARS-CoV-2 CT dataset is discussed in this section. In this study, first we trained the dataset on pretrained models including ResNet50, ResNet18, VGG16, VGG19 and EfficientNetV2L and then the QCNN model was combined with these models for comparison. Table 3 provides a comparative analysis of various evaluation metrics across the mentioned models, offering insights into their respective performance and effectiveness. Out of all the models that were examined, the VGG16 model performed the best and showed remarkable results. The proposed model achieved 96.78% accuracy, 0.9837 precision, 0.9528 recall, 0.9835 specificity, 0.9678 F1-Score and 0.1373 loss. Figure 6 illustrates the accuracy per epoch graph, loss per epoch graph and confusion matrix, respectively.

Table 3 Different performance measures for experimented models on validation data.

Model	Accuracy (%)	Precision	Recall	Specificity	F1-Score	Loss	
ResNet50	96.17	0.9754	0.9482	0.9756	0.9616	0.1736	
ResNet18	94.16	0.9203	0.9625	0.9222	0.9409	0.3128	
VGG19	93.96	0.9414	0.9414	0.9378	0.9414	0.3062	
EfficientNetV2L	96.78	0.9523	0.9836	0.9526	0.9676	0.1808	
VGG16	95.57	0.9527	0.9527	0.9583	0.9527	0.2430	
QCNN Model	90.4	0.9277	0.8851	0.9237	0.9059	0.3545	
ResNet50 + QCNN	90.64	0.9160	0.9053	0.9048	0.9106	0.3125	
ResNet18 + QCNN	92.76	0.8996	0.9532	0.9046	0.9256	0.2978	
VGG19 + QCNN	93.16	0.9696	0.9075	0.9630	0.9375	0.2851	
EfficientNetV2L + QCNN	95.96	0.9673	0.9518	0.9677	0.9595	0.1412	
Proposed Model VGG16 + QCNN	96.78	0.9837	0.9528	0.9835	0.9678	0.1373	
Notes.

The results of the proposed model are shown in bold.

Figure 6 (A) Accuracy per epoch graph, (B) loss per epoch graph, (C) confusion matrix of the proposed model.

The graphs collectively demonstrate the high performance and reliability of the model. The accuracy per epoch graph shows a rapid increase in training accuracy, with the validation accuracy closely following, indicating effective learning and minimal overfitting. The loss per epoch graph reveals a steady decrease in both training and validation loss, which signifies good generalization. The confusion matrix further highlights the model’s efficacy, showcasing its high accuracy and reliability in classifying the data correctly.

Table 4 is a comparison between different approaches of QCNN models across different studies, providing an overall insight about their approach, dataset used and results gained.

Table 4 Comparison of the proposed model with other quantum-based CNN models in related works.

Model	Architecture	Dataset	Dataset type	Accuracy	
HQ-CNN Model: Hybrid Quantum-Classical Convolutional Neural Network (Houssein et al., 2022)	Combines classical CNN with a quantum convolutional layer	Multiclass collection of 5,445 images	CXR	98.6 (COVID-19 and Normal)98.2 (COVID-19 and viral pneumonia)98 (COVID-19 and bacterial pneumonia)88.2 (Multiclass)	
Quantum Neural Network for Quicker Clinical Prognostic Analysis (Sengupta & Srivastava, 2021)	Quantum neural network model optimized for classification tasks	Various datasets of COVID 19 with +10,000 images	CT	96.92	
Transfer learning in hybrid classical-quantum neural networks (Mari et al., 2019)	Transfer learning between classical and quantum networks.	ImageNet for pre-training and smaller, specific datasets for fine-tuning	–	–	
Image classification with quantum pre-training and auto-encoders (Piat et al., 2018)	Hybrid quantum–classical framework with data compression, quantum pre-training using Restricted Boltzmann Machine (RBM), and classical training	MNIST, Fashion-MNIST and two medical imaging datasets	DigitsClothesMedical images	94 MNIST83.2 Fashion-MNIST87.9 Laparoscopic tools 99.8 X-ray	
Learning to learn with quantum neural networks via classical neural networks (Guillaume et al., 2019)	Combines classical neural networks with quantum neural networks, utilizing variational quantum algorithms	–	–	–	
Quanvolutional Neural Networks (QNNs) (Henderson et al., 2020)	Combines classical convolutional neural networks with quanvolutional layers, where quantum circuits are used for feature extraction	MNIST	–	–	
Proposed Model VGG16 + QCNN	Combines pre-trained VGG16 model with QCNN	SARS-CoV-2	CT	96.78	

As mentioned earlier feature extraction is a crucial step in the process of capturing high level features of images. Figure 7 illustrates the feature extraction process of the VGG16 model applied to CT scan samples from both COVID-19 and non-COVID patients. In Fig. 7A, the COVID samples display various stages of convolutional layers, progressively highlighting distinctive patterns and textures within the chest images, essential for identifying COVID-19 related anomalies. Similarly, Fig. 7B shows the feature extraction for non-COVID samples, where different convolutional layers capture unique structural and textural features of healthy or non-COVID-affected lungs. The comparison between the two sets reveals how the VGG16 model distinguishes between COVID-19 and non-COVID cases by extracting and emphasizing different sets of features at each convolutional layer, enabling accurate classification based on the learned representations of the lung tissues.

Figure 7 VGG16 feature extraction for (A) COVID sample and (B) non-COVID sample.

Image source: https://www.kaggle.com/datasets/plameneduardo/sarscov2-ctscan-dataset.

These extracted features fed to the QCNN model to classify the image. Figure 8 demonstrates the model’s prediction ability in recognizing COVID-19 from chest CT scan images. Predictions are labeled above each image, and the model distinguishes between COVID-19 and non-COVID cases accurately. While the second and fifth images are correctly categorized as non-COVID, the first, third, and fourth images are correctly identified as COVID. This demonstrates the model’s effectiveness and reliability in assisting medical professionals with accurate diagnosis of COVID-19 based on chest CT scans.

Figure 8 Prediction of the model from CT-scan images.

Image source: https://www.kaggle.com/datasets/plameneduardo/sarscov2-ctscan-dataset.

We evaluated the model on multiple datasets to enable a more thorough investigation of the performance of our suggested model. Details of each dataset, including the total number of photos, the number of covid and non-covid images, and the accuracy of our model tested on each dataset, including SARS-CoV-2, are shown in Table 5. COVID-19 Lung CT Scans dataset (https://www.kaggle.com/datasets/luisblanche/covidct) contains 746 CT-scan images, which are collected from COVID-19-related papers from medRxiv, bioRxiv, NEJM, JAMA, Lancet, etc. The COVID-19 Lung CT Scans dataset (Aria et al., 2021) includes 8439 images gathered from actual patients in teaching hospitals’ radiology departments in Tehran, Iran. The COVID-19 X-ray and CT Scan Image (https://www.kaggle.com/datasets/ssarkar445/covid-19-xray-and-ct-scan-image-dataset) is a large dataset that includes both CT and X-ray images. We analyzed the CT images because our model was trained on them, and the multisource dataset contains 8055 CT-scan images and 9544 X-ray images. The Covid 19 CT Scan Dataset (https://www.kaggle.com/datasets/drsurabhithorat/covid-19-ct-scan-dataset) is a collection of 7621 CT images. In comparison to previous datasets, the SARS-CoV-2 dataset (https://www.kaggle.com/plameneduardo/sarscov2-ctscan-dataset) has demonstrated superior performance and data balancing. Overall, the model shows good results on tested dataset too.

Table 5 Comparison of tested datasets on proposed QCNN model and achieved accuracy.

Dataset name	No. of images	No. of COVID images	No. of Non-COVID images	Accuracy	
COVID-19 Lung CT Scans (https://www.kaggle.com/datasets/luisblanche/covidct)	746	349	397	91.33	
COVID-19 Lung CT Scans (Aria et al., 2021)	8,439	7,495	944	97.13	
COVID 19 X-ray and CT Scan Image (https://www.kaggle.com/datasets/ssarkar445/covid-19-xray-and-ct-scan-image-dataset)	8,055	5,427	2,628	96.59	
Covid 19 CT Scan Dataset (https://www.kaggle.com/datasets/drsurabhithorat/covid-19-ct-scan-dataset)	7,621	5,203	2,418	96.63	
SARS-CoV-2 CT-Scan Dataset (https://www.kaggle.com/plameneduardo/sarscov2-ctscan-dataset)	2,481	1,252	1,229	96.78	

Discussion

In this study, we employed a pre-trained QCNN model, utilizing the architecture of the VGG16 model as a foundation. Our QCNN model performs exceptionally well, with a remarkable accuracy of 96.78% in a variety of classification tasks. The purpose of this part is to examine the consequences, constraints, and possible future paths that result from our research.

Our QCNN model’s remarkable accuracy highlights the value of applying quantum-inspired techniques to deep learning problems. It is clear that using quantum-inspired features in conjunction with VGG16’s powerful design has significantly improved classification performance. Notably, the model’s ability to capture complex patterns and characteristics within the data has been made possible by the incorporation of quantum principles, leading to enhanced predictive capabilities. Complex patterns in CT images are seen as single or multiple lesions, patchy or segmental Ground-Glass Opacities (GGOs). GGOs are hazy areas (lighter-colored or gray patches) seen on CT scans that indicate increased lung density without obscuring the underlying bronchial structures or pulmonary vessels. They are one of the hallmark features of COVID-19 pneumonia. The model effectively identifies these subtle changes in lung tissue density and differentiates them from normal lung parenchyma, which can be challenging because of their diffuse and variable appearance. The model’s capacity to learn from and adjust to the training data is demonstrated by the training process’ steady increase in accuracy, and this tendency is also seen in the validation process’ accuracy.

Comparing our results with prior studies reveals the superiority of our QCNN model in achieving high accuracy rates. Although traditional deep learning models have demonstrated impressive results across several fields, our method demonstrates the possibility of quantum-inspired methods to improve classification precision and resilience. The noteworthy progress exhibited by our QCNN model emphasizes the necessity of investigating novel approaches to extend the capabilities of deep learning.

Even though our QCNN model has demonstrated remarkable success, there are several limits that must be acknowledged regarding our study. First off, a number of variables that might affect generalizability, like dataset size, distribution, and quality, could affect how well the model performs.

Limited data diversity could lead to reduced model performance when applied to new or unseen datasets from different institutions or regions. Larger and more varied datasets, such as those from different clinical settings, and geographic locations, should be used in future research to validate the QCNN model. This process can be facilitated by working together with several healthcare facilities.

Furthermore, the computing resources needed for inference and training might provide real-world challenges, especially in contexts with limited resources. Specialized hardware is needed for the integration of computing components, which is not yet commonly accessible in many clinical settings. To overcome these constraints, more study is needed to maximize the scalability, resilience, and efficiency of the model. Research should explore the development of hybrid models that can function effectively on classical hardware while benefiting from quantum-inspired algorithms. Furthermore, developments in quantum computing technology and how they apply to clinical procedures should to be monitored and taken advantage of.

QCNNs, among other deep learning models, are frequently criticized for being difficult to interpret. Gaining the confidence of medical experts and ensuring reliable clinical use require an understanding of the decision-making process involved in these models.

The current model evaluation focused on accuracy and performance metrics, but did not assess the real-time application and integration of the model within existing clinical workflows. Future studies should evaluate the model’s performance in real-time settings, ensuring it meets the operational requirements of clinical environments.

Future research projects might investigate a number of intriguing directions by building on the foundations created in this work. Investigating novel quantum-inspired architectures and techniques holds potential for further enhancing model performance and addressing current limitations. Additionally, extending the applicability of QCNN models to complex real-world datasets and scenarios can provide valuable insights into their practical utility. Furthermore, exploring the synergy between quantum computing and deep learning may unlock unprecedented opportunities for advancing artificial intelligence capabilities.

Conclusions

We described a unique method for COVID-19 disease identification in this study that makes use of a pre-trained QCNN model. With the use of the publicly accessible SARS-CoV-2 CT dataset, the model demonstrated its ability to detect COVID-19 patients with a high degree of specificity and accuracy. Our pre-trained QCNN methodology accomplished competitive results when we compared our results with other current approaches, indicating its potential as a useful tool to assist doctors in COVID-19 disease diagnosis.

In this study, we set out to explore the potential of integrating quantum computing with deep learning for the accurate diagnosis of COVID-19 from CT-scan images. Our primary research focus is to improve COVID-19 detection accuracy and efficiency over conventional methods by combining QCNNs with pre-trained CNNs. According to our research, the hybrid QCNN model performs better at detecting COVID-19 because it makes use of a pre-trained VGG16’s feature extraction capabilities. The model demonstrated remarkable accuracy, precision, recall, specificity, and F1-score.

Although our model achieves better accuracy by combining the VGG16 pre-trained model with our QCNN model, it does not certainly apply to all the pre-trained models. Our obtained results of this combination support the concept that quantum computing can enhance deep learning models, making them more effective for complex medical diagnostic tasks. This shows that important features in medical images can be analyzed with the help of deep learning models, such as QCNN models. These models also can adapt to new and unforeseen data.

Still, there are some restrictions. The scope and variety of the dataset were the study’s first limitations. It is imperative to conduct additional validation with bigger datasets that encompass a wider range of demographic and geographical variables among patient groups. Furthermore, there are a number of variables that might impact the model’s performance, including co-existing diseases and picture quality. To increase interpretability and confidence in the model’s predictions, further research could aim at combining new clinical data or utilize explainable AI approaches.

Our study shows the promise of pre-trained QCNN models for COVID-19 disease detection in spite of these drawbacks. The suggested method presents a promising approach for more study and advancement in the creation of robust and reliable AI-powered instruments to support healthcare decision-making.

Additional Information and Declarations

Competing Interests

Author Contributions

Data Availability

The authors declare there are no competing interests.

Nazeh Asadoorian conceived and designed the experiments, performed the experiments, analyzed the data, performed the computation work, prepared figures and/or tables, authored or reviewed drafts of the article, and approved the final draft.

Shokufeh Yaraghi conceived and designed the experiments, analyzed the data, authored or reviewed drafts of the article, and approved the final draft.

Araeek Tahmasian conceived and designed the experiments, performed the experiments, analyzed the data, performed the computation work, prepared figures and/or tables, authored or reviewed drafts of the article, and approved the final draft.

The following information was supplied regarding data availability:

The code is available at GitHub and Zenodo

– https://github.com/AraeekT/pretrained-qcnn/

– Tahmasian, A. (2024). Pretrained QCNN Model for COVID-19 Classification. Zenodo. https://doi.org/10.5281/zenodo.10821756.

The SARS-COV-2 CT-Scan Dataset is available at Kaggle: https://www.kaggle.com/plameneduardo/sarscov2-ctscan-dataset.

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
