# Peer review of "Pre-trained quantum convolutional neural network for COVID-19 disease classification using computed tomography images"

_PeerJ Computer Science, doi:10.7717/peerj-cs.2343_

## Round 0.1 · original submission · Major Revisions

Dear Authors,

Your paper has been reviewed. Based on the reviewers' reports, major revisions are needed before it is considered for publication in PeerJ Computer Science.

More precisely, the following issues must be solved by the authors:

1. The impact and novelty of the findings must be better asses. While the replication of results is encouraged, the paper should also highlight the novel contributions more distinctly. Furthermore, the manuscript would benefit from more detailed descriptions of the figures to enhance clarity and impact.
2. Access to raw data must be granted if possible. This point is crucial for reproducibility.
3. While the paper's experimental design is robust, the description of the methods must be improved by specifying the software, tools, or specific versions used in the study to enhance reproducibility.
4. Conclusions must be expanded to more explicitly link back to the original research questions and highlight how the results support or advance previous findings.
5. Finally, a flowchart for the proposed QCNN model must be included and explained.

To conclude, I advise the authors to consider all the reviewers' comments before resubmitting their revised paper.

Reviewer 1 ·

Basic reporting

Strengths:

1. The paper is well-structured, adhering to the required standards with clear, professional English throughout.
2. The introduction provides an adequate background, placing the research in context with well-referenced and relevant literature.
3. Figures are relevant and generally of high quality, with appropriate labeling and descriptions.

Weaknesses:

1. The manuscript would benefit from more detailed descriptions of the figures to enhance clarity and impact.
2. It's unclear if the raw data is fully supplied as per the PeerJ policy. Ensuring access to raw data is crucial for reproducibility.

Experimental design

Strengths:

1. The research question is well defined and highly relevant, aiming to fill a clear knowledge gap within the scope of the journal.
2. Methods are described with sufficient detail, allowing for potential replication of the study.
3. The study adheres to high ethical standards with a rigorous investigation process.

Weaknesses:

1. While the experimental design is robust, the description of the methods could be improved by specifying the software, tools, or specific versions used in the study to enhance reproducibility.
2. Conclusions are well-stated but could be expanded to more explicitly link back to the original research questions and highlight how the results support or advance previous findings.

Validity of the findings

Strengths:

1. The findings are robust and statistically sound, with clear evidence of data control.
2. The paper demonstrates a meaningful advancement in the use of quantum convolutional neural networks for COVID-19 detection using CT images, potentially filling a significant gap in rapid diagnostic methods.

Weaknesses:

1. The paper does not assess the impact and novelty of the findings sufficiently. While the replication of results is encouraged, the paper should also highlight the novel contributions more distinctly.
2. More detailed discussion on the limitations of the findings and how they relate to the realistic application in clinical settings would be beneficial.

Additional comments

The paper is a significant contribution to the field of medical imaging and AI, particularly in the context of the COVID-19 pandemic. It leverages advanced technologies in a meaningful way. However, enhancing the clarity of raw data availability, expanding methodological descriptions, and a stronger focus on novelty and impact could improve the manuscript significantly.

·

Basic reporting

In this manuscript author proposed a pre-trained quantum convolutional neural network (QCNN) for rapid detection of Covid-19. The pre-trained QCNN model combines the strength of quantum computing with the feature extraction capabilities of VGG16. Results of proposed QCNN model is compared with other pre-trained models such as ResNet50, ResNet18, VGG19, and EfficientNetV2L. But authors not discussed about the significance of proposed QCNN model compared to other existing quantum based CNN models such as Hybrid quantum-classical convolutional neural network model for COVID-19 prediction [32], Quantum algorithm [33], HQDCNet [34], hybrid classical-quantum neural networks [35], quantum pre-training and autoencoders [36], quantum neural networks [37], Quanvolutional neural networks [38],etc.
In abstract author can state how the proposed QCNN model is better than other existing quantum based CNN models that are used in detection of Covid-19. What is the significance of the proposed QCNN compared to other quantum based CNN models.
In literature review authors can state the identified research gap. Why the pertained CNN models, ResNet50, ResNet18, VGG19, and EfficientNetV2L not able to identify complex patterns and characteristics within COVID-19 images and how quantum components helps in identify the complex patterns and characteristics within COVID-19 images.

Experimental design

Authors have compared the results of QCNN with other pre-trained models such as ResNet50, ResNet18, VGG19, and EfficientNetV2L. Authors can also include comparison of the results of QCNN with other quantum based CNN models.

Authors conducted experiments by training and testing QCNN model using dataset collected from single geographical locations (SARS-CoV-2 dataset of COVID-19 CT-scan images collected from 313 hospitals in Sao Paulo, Brazil).Authors can test the pre-trained QCNN model using multisource dataset, COVID-19 CT-scan images collected from other geographical locations that are available in kaggel and the results can be included in tables and graphs. Authors can discuss how the pre-trained QCNN model performs when tested with images from same dataset and multisource dataset.

In materials and methods section, under the Architecture of QCNN, authors can include details on the role of Estimator QNN, ZZfeaturemaps and Real Amplitudes in improving the model capacity.

For better understanding flowchart for the proposed QCNN model can be included and explained.

Validity of the findings

In the proposed QCNN model, author stated that VGG16 and quantum principles are incorporated to enhance prediction accuracy of the model by capturing complex patterns and characteristics within the images. Authors can explain what are complex patters and characters that are captured by the proposed QCNN model. For better understanding include figures that represents features extracted images at various layers of VGG16 model (without Quantum Components), QCNN model and the results can be compared. Authors can include a figure with samples of normal and COVID-19 images processed by QCNN model at various layers and can be explained in detail.

Additional comments

Nil

---

## Round 0.2 · accepted · Accept

Dear Authors,

Your paper has been accepted for publication in PEERJ Computer Science. The comments of the reviewers who evaluated your manuscript are included in this letter. I ask that you make minor changes to your manuscript based on those comments, before uploading final files. Thank you for your fine contribution.

·

Basic reporting

Author have done significant corrections as per the reviewers comments. Additional results are included in Table 3 an Table 4 are appreciable. Discussion on significance of proposed QCNN model compared with other quantum based CNN models is convincing. Research gaps are identified and the authors have explained well how the proposed model overcomes the identified research gaps .

Experimental design

Author have done significant corrections as per the reviewers comments. Inclusion of Table 5 in the manuscript is appreciable and discussions related to Table 5 is convincing. Inclusion of figure 4 (flowchart ) is appreciable and discussions related to it is convincing.

Validity of the findings

Author have done significant corrections as per the reviewers comments. Inclusion of Figure 7 to show feature extractions and Figure 8 to demonstrate output prediction have added values to the manuscript and the discussions are convincing.

Additional comments

Inclusion of Table 4, Table 5 , Figure 7 and Figure 8 and the discussions related to it have added values to the manuscript.